# Isolated copper–tin atomic interfaces tuning electrocatalytic CO$_2$ conversion

Wenhao Ren[1,7], Xin Tan[2,7], Jiangtao Qu [3,4], Sesi Li[5], Jiantao Li [6], Xin Liu[5], Simon P. Ringer [3], Julie M. Cairney [3,4], Kaixue Wang [5], Sean C. Smith[2] & Chuan Zhao [1✉]

Direct experimental observations of the interface structure can provide vital insights into heterogeneous catalysis. Examples of interface design based on single atom and surface science are, however, extremely rare. Here, we report Cu–Sn single-atom surface alloys, where isolated Sn sites with high surface densities (up to 8%) are anchored on the Cu host, for efficient electrocatalytic CO$_2$ reduction. The unique geometric and electronic structure of the Cu–Sn surface alloys (Cu$_{97}$Sn$_3$ and Cu$_{99}$Sn$_1$) enables distinct catalytic selectivity from pure Cu$_{100}$ and Cu$_{70}$Sn$_{30}$ bulk alloy. The Cu$_{97}$Sn$_3$ catalyst achieves a CO Faradaic efficiency of 98% at a tiny overpotential of 30 mV in an alkaline flow cell, where a high CO current density of 100 mA cm$^{-2}$ is obtained at an overpotential of 340 mV. Density functional theory simulation reveals that it is not only the elemental composition that dictates the electrocatalytic reactivity of Cu–Sn alloys; the local coordination environment of atomically dispersed, isolated Cu–Sn bonding plays the most critical role.

[1] School of Chemistry, University of New South Wales, Sydney, NSW, Australia. [2] Integrated Materials Design Laboratory, Department of Applied Mathematics, Research School of Physics, The Australian National University Canberra, Canberra, ACT, Australia. [3] Aerospace, Mechanical and Mechatronic Engineering, The University of Sydney, Sydney, NSW, Australia. [4] Australian Centre for Microscopy and Microanalysis, The University of Sydney, Sydney, NSW, Australia. [5] School of Chemistry and Chemical Engineering, Shanghai Jiao Tong University, Shanghai, China. [6] State Key Laboratory of Advanced Technology for Materials Synthesis and Processing, Wuhan University of Technology, Wuhan, China. [7] These authors contributed equally: Wenhao Ren, Xin Tan. ✉email: chuan.zhao@unsw.edu.au

Electrocatalytic $CO_2$ reduction reaction ($CO_2$RR) offers a sustainable approach to convert the greenhouse gas into value-added chemicals and fuels, and in the meantime store the intermittent renewable electricity[1–4]. The control over the selectivity of $CO_2$RR is a major challenge given that it can produce up to 16 different gas and/or liquid products depending on diverse reaction pathways[5–8], while competing with hydrogen evolution reaction (HER) that is thermodynamically more favorable[9,10]. To achieve efficient $CO_2$RR with targeted products, the design and synthesis of catalysts with optimal bonding (not-too-strong, not-too-weak) to key intermediates are essential[11]. However, the mean-field behavior typically dominates the adsorption properties of heterogeneous catalysts owing to their broad electronic band structures[12]. As a result, the current metal-based catalysts generally follow linear scaling relationships[13,14], which place fundamental limitations to catalytic performance in terms of reactivity and selectivity.

Alloy design is one promising approach towards effective tailoring of the metal catalyst's properties. An important merit of polymetallic materials is that the interactions between different metal atoms at the interfaces can result in significantly enhanced catalytic activity owing to both geometric and electronic effects[15–18]. Whereas, alloys generally follow scaling relationships[19]. It has been predicted that the microalloying could induce band narrowing and adsorption properties that cannot be calculated from a linear interpolation, owing to the isolated single-metal-site bonding environment[12,20,21]. For example, surface alloys that comprise catalytically active single atoms (e.g., Au, Pt, Pd, etc.) anchored on the host surface have shown exceptional properties for various catalytic applications[22–25]. With this design, no bonds form between foreign active sites; thus the intermetallic interactions occurring at the interfaces between two metals are fundamentally different from bulk alloys[26]. Meanwhile, their single-site nature can support the atom utilization up to 100%, which could substantially reduce the cost and tackle the scarcity issues of using precious elements. Besides, the well-defined active sites in single-atom surface alloys can provide an ideal model for mechanistic studies and enable the rational construction of catalysts through a three-pronged approach, comprising theoretical modeling, surface science, and catalyst assessment under industrially relevant conditions.

Herein, we demonstrate Cu–Sn surface alloys ($Cu_{97}Sn_3$ and $Cu_{99}Sn_1$) with isolated Sn atoms on Cu host surface by a simple sequential reduction process. Previous studies have shown that Cu–Sn alloys can realize both $CO_2$-to-CO[27,28] and $CO_2$-to-formate reaction pathways[29,30] (see Supplementary Table 2). Although insights have been gained in terms of morphology[31], structure[30], valence state[32], and composition[29], the mechanistic understanding of Cu–Sn alloys at the isolated atomic interfaces and the root behind different $CO_2$RR pathways remain elusive. In this study, the three-dimensional atomic structure of the Cu–Sn surface alloy is established by using the atom probe tomography (APT) and high-angle annular dark-field scanning transmission electron microscopy (HAADF-STEM). The $Cu_{97}Sn_3$ surface alloy exhibits near-unity selectivity for $CO_2$-to-CO conversion, which is completely distinct from pure Cu and Cu–Sn bulk alloys. Density functional theory (DFT) simulations reveal the crucial role of the isolated Cu–Sn alloy-bonding geometry, which shows optimal absorption property of intermediates (COOH* and CO*) compared with pure Cu (too strong) or $Cu_{70}Sn_{30}$ bulk alloy (too weak).

## Results

### Synthesis and structural characterization.
A family of Cu–Sn nanoparticles with an average size of 15 nm featuring different surface structures, were synthesized by a single-step reduction process using $Cu^{2+}$, $Sn^{2+}$, and $NaBH_4$ (Fig. 1a). By controlling the

content of Sn, single-atom surface alloys (e.g., $Cu_{99}Sn_1$ and $Cu_{97}Sn_3$) and Cu@Sn core-shell bulk alloy ($Cu_{70}Sn_{30}$) can be obtained, respectively. The ability to fabricate these complex structures in a single reaction step primarily stem from the sequential reduction processes, owing to the more negative standard electrode potential of $Sn^{2+}$ relative to $Cu^{2+}$ ($Sn^{2+} + 2e^- \rightleftharpoons$ Sn, $E^0 = -0.13$ V vs. SHE; $Cu^{2+} + 2e^- \rightleftharpoons$ Cu, $E^0 = 0.34$ V vs. SHE). As a result, the reduction of $Cu^{2+}$ takes place first, followed by the deposition of Sn on Cu surface (or near the surface). The elemental distributions of $Cu_{97}Sn_3$ and $Cu_{70}Sn_{30}$ are shown in energy-dispersive X-ray spectroscopy mapping (Fig. 1b, c and Supplementary Fig. 2), where the surface segregation of Sn can be clearly observed. This is consistent with the HAADF-STEM that the Sn atoms preferentially anchored at the outer shell of Cu nanoparticles (Fig. 1d–f). The atomic dispersion of Sn in $Cu_{97}Sn_3$ can be further identified from the distributed bright spots in the HAADF-STEM image (Fig. 1e and Supplementary Fig. 3), which is in sharp contrast to the clean crystal of $Cu_{100}$ (Fig. 1d). The detailed synthetic procedure for a set of Cu, $Cu_{99}Sn_1$, $Cu_{97}Sn_3$, and $Cu_{70}Sn_{30}$ samples is provided in the experimental section.

Atom probe tomography (APT) is a powerful tool to reveal the atomic configuration of catalysts by virtue of capabilities in chemical composition measurements and three-dimensional (3D) imaging at the atomic level[33]. APT experiments normally are terminated as a result of tip rupture due to electrical force, and therefore a smooth and compact tip has been prepared by delicate focused ion beam (FIB) annular milling to improve the tip survival during the experiment (Fig. 2a). The corresponding 3D tomography of this tip is displayed in Fig. 2b, where the content of Cu is more than an order of magnitude higher than Sn as shown in the full range mass-to-charge spectrum (Supplementary Fig. 4e), consistent with our experimental design. The Sn atoms (green dots) are thus shown in a larger size than Cu atom (purple dots) to highlight its position. Notably, the mass spectrum of Sn matches very well to the corresponding natural abundance of main isotopes (blue lines), which is a strong evidence that these peaks are not from impurities or noise (Supplementary Fig. 4f). The isolation of Sn atoms on Cu matrix can be well-distinguished from an animation (Supplementary Movie 1). In addition, we created two Sn iso-surfaces with densities of 3.53 wt.% (Fig. 2c) and 0.63 wt.% (Fig. 2d), respectively, to investigate the Sn-rich and Sn-lean region on the Cu host. The tomography demonstrates a non-uniform distribution of Sn dopants, with more Sn atoms detected on the left side. Moreover, the proxigram profile was also created based on the Sn iso-surface of 3.53 wt.% (Fig. 2e), indicating the compositional profile of Sn with respect to the distance from this iso-surface (the zero-coordinate origin of proxigram). The profile shows an even higher Sn content up to ~8% on the left side of this Sn-rich surface, which is in line with our statistic binomial analysis of Sn distribution in Fig. 2f. Besides, we projected the entire Sn atoms on $xy$ plane (perpendicular to $z$-direction) by creating a 2D Sn density contour map (Fig. 2g). It is intriguing to observe that the Sn-rich region exhibits an arc-shape and a distinguishable interface with the Sn-lean region. Note that APT detects only half of the $Cu_{97}Sn_3$ nanoparticle, with the other half at the tip of the needle apex possibly milled away by FIB annular milling. The arc-shape interface thus can be related to the surface of Cu particles, which is doped with isolated Sn atoms. These APT results are in agreement with our surface alloy design, namely high surface densities of isolated Sn atoms are formed on the surface of Cu nanoparticles to create exposed Cu–Sn atomic interfaces.

X-ray diffraction (XRD) pattern of the catalysts is shown in Supplementary Fig. 5a, where the $Cu_{100}$ and $Cu_{97}Sn_3$ exhibit identical diffraction peaks arising from the *fcc* structure of Cu and $Cu_2O$, indicative of the atomic dispersion of Sn without alloy phase formed. The formation of $Cu_2O$ can be attributed to the

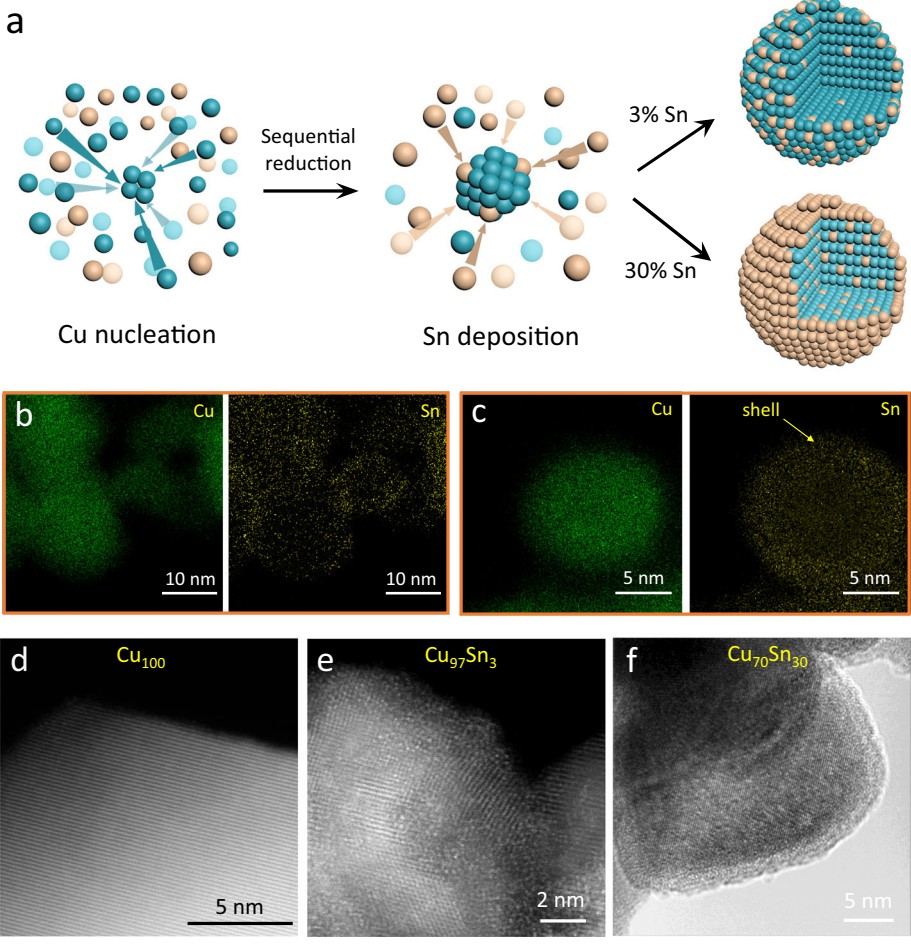

**Fig. 1 Synthesis and structural characterization. a** Schematic illustration of the Cu–Sn nanoparticle formation via the sequential reduction. STEM/EDX mapping images of $Cu_{97}Sn_3$ (**b**) and $Cu_{70}Sn_{30}$ (**c**). HAADF-STEM image of $Cu_{100}$ (**d**), $Cu_{97}Sn_3$ (**e**), and $Cu_{70}Sn_{30}$ (**f**).

spontaneous oxidation of Cu nanoparticle surface when exposed to the air. Note that there is no $Cu_2O$ diffraction peak observed in $Cu_{70}Sn_{30}$, possibly due to the coating of Sn layer that protects Cu from oxidation. This is also supported by the UV–vis spectroscopy (Supplementary Fig. 5b), where the coating of Sn layer on $Cu_{70}Sn_{30}$ also blocks the plasmonic signal of Cu, leading to the absence of the absorption peak at ~620 nm. As a result, the valence states of Cu on the surfaces of $Cu_{100}/Cu_{97}Sn_3$ and $Cu_{70}Sn_{30}$ are estimated to be +1 and 0, respectively, as shown in Cu LMM Auger spectra (Supplementary Fig. 5c).

The Cu K edge X-ray absorption near-edge structure (XANES) and extended X-ray absorption fine structure (EXAFS) measurements were further carried out to uncover the coordination environment and electronic structure of samples. Cu foil and CuO were used as references for metallic Cu and $Cu^{2+}$, respectively. The near-edge positions of $Cu_{100}$, $Cu_{97}Sn_3$, and $Cu_{70}Sn_{30}$ catalysts are slightly higher than that of Cu foil, indicating that the bulk Cu in the three samples is in the metallic state with only surface been oxidized to $Cu^+$ (Supplementary Fig. 6a)[34]. The Fourier-transformed $k^3$-weighted spectra of the samples exhibit two major peaks at 1.5 and 2.2 Å, corresponding to Cu–O and Cu–Cu coordination shells, respectively (Supplementary Fig. 6b). The Cu–O peak is only observed in $Cu_{97}Sn_3$ and $Cu_{100}$ catalysts, while the $Cu_{70}Sn_{30}$ displays similar peak shape compared to Cu foil with peak intensity from 4 to 5 Å significantly suppressed. This result also suggests that the surface of $Cu_{70}Sn_{30}$ was completely coated with Sn shell.

**Evaluation of CO₂RR performances**. The selectivity, activity, and stability of $Cu_{100}$, $Cu_{97}Sn_3$, and $Cu_{70}Sn_{30}$ for CO₂RR were investigated in a CO₂-saturated 0.5 M KHCO₃ electrolyte (pH = 7.2). The gas and liquid products were quantified by gas chromatography (GC) and nuclear magnetic resonance (NMR), respectively (Supplementary Fig. 7). It is worth noting that Cu–Sn alloys could be oxidized when exposed to the air. Therefore, a cyclic voltammetry (CV) pretreatment was conducted in relatively negative potentials from −0.5 to −2 V vs. RHE for ten cycles prior to CO₂RR, to remove the possible surface oxides and exclude their influence on CO₂RR (Supplementary Fig. 8). As confirmed by the ex situ Sn 3$d$ XPS and Cu LMM Auger spectra, most of the oxides on $Cu_{97}Sn_3$ were reduced after the CV pretreatment (Supplementary Fig. 9). Figure 3a–c show potential-dependent Faradaic efficiencies (FEs) of different products obtained with $Cu_{100}$, $Cu_{97}Sn_3$, and $Cu_{70}Sn_{30}$ catalysts. The pure $Cu_{100}$ exhibits poor selectivity for CO₂RR with diverse products including $H_2$, CO, formate, and $C_2H_4$, in agreement with previous reports on Cu-based catalysts[35,36]. Surprisingly, with only 3% of Sn single-atoms alloyed on Cu surface, the selectivity completely changed to CO₂-to-CO conversion. The onset potential obtained with the $Cu_{97}Sn_3$ catalyst was as low as −0.4 V, and the maximum $FE_{CO}$ of 98% was achieved at −0.7 V. To further verify the surface alloy effect, the $Cu_{99}Sn_1$ was synthesized and evaluated (Supplementary Fig. 11). It is found that even with only 1% of Sn, the Cu catalyst can yield a high selectivity of ~90% for CO production. In comparison,

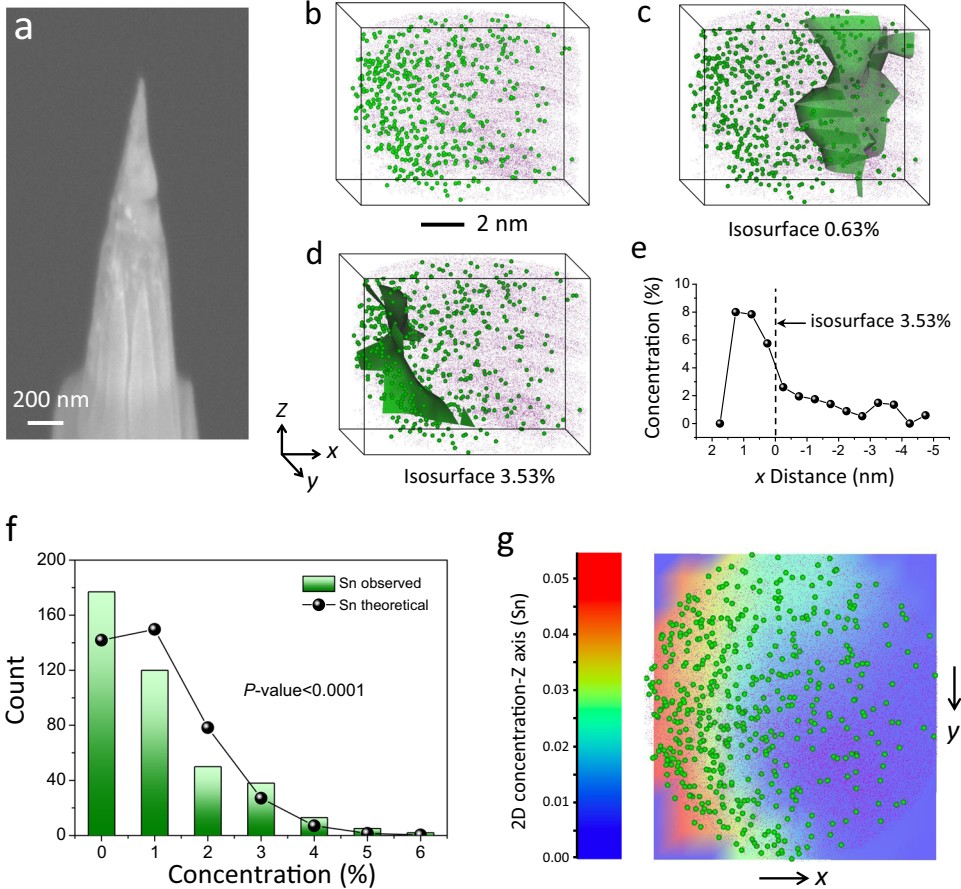

**Fig. 2 Atom probe tomography. a** SEM side view (54° tilted) of the needle-shape specimen. **b** 3D tomography of the $Cu_{97}Sn_3$ nanoparticle demonstrates the position relationship of Sn dopants (green) and Cu matrix (purple). **c** Sn iso-surface of 0.63 wt.% indicates the low dopant concentration region on the Cu nanoparticle, in contrast to (**d**) the Sn-rich region evidenced by the 3.53 wt.% Sn iso-surface. The z-direction of tomography corresponds to the tip axial direction. **e** A proxigram profile was in accordance with 3.53 wt.% Sn iso-surface, demonstrating the Sn composition reduces along the arrow direction in (**d**). **f** Frequency distribution analysis confirms a non-random distribution of Sn dopants. **g** The Sn concentration 2D contour projected on xy plane indicates a curved Sn-rich band, which corresponds to the Cu particle surface, and the low dopant region refers to the inside of the Cu particle.

when the content of Sn increases to 30% to form the $Cu_{70}Sn_{30}$ bulk alloy, the catalyst selectivity is switched to formate production (FE of ~90% at −1.1 V) with $CO_2$-to-CO conversion almost suppressed. The product distribution obtained with $Cu_{70}Sn_{30}$ catalyst is similar to those for the Sn[37], SnO[38], and several Cu–Sn alloy electrodes[29].

As shown in linear sweep voltammetry (LSV) curves (Fig. 3d), the total current densities of $Cu_{97}Sn_3$ outperform both $Cu_{100}$ and $Cu_{70}Sn_{30}$ from −0.4 to −1.0 V vs. RHE, reaching a current density as high as 112 mA cm$^{-2}$ at −1.0 V. Note that the onset potential of $Cu_{97}Sn_3$ decreases for ~230 mV compared to $Cu_{70}Sn_{30}$, which can be attributed to different reaction pathways and will be further discussed below. Figure 3e shows Sn-mass-normalized $CO_2RR$ partial currents obtained with $Cu_{97}Sn_3$ and $Cu_{70}Sn_{30}$. A high current density of 2.3 A mg$_{Sn}^{-1}$ was achieved on $Cu_{97}Sn_3$ at −0.8 V, a value 23 times greater than that for $Cu_{70}Sn_{30}$. For the stability tests (Fig. 3f), the $Cu_{97}Sn_3$ exhibits a robust response with 97% retention of $j_{CO}$ and almost unchanged $FE_{CO}$, even after 20 h of consecutive electrolysis at high current densities of ~30 mA cm$^{-2}$. The electrochemically active surface area (ECSA) is evaluated from double-layer capacitance ($C_{dl}$) (Supplementary Fig. S13) to elucidate the origin of diverse catalytic activity of the electrodes[39,40]. The $Cu_{100}$, $Cu_{97}Sn_3$, and $Cu_{70}Sn_{30}$ show similar $C_{dl}$ (2.0–2.5 mF cm$^{-2}$) and ECSA, suggesting that the observed catalytic property is primarily

contributed by the intrinsic reactivity of each active site, rather than a surface area effect.

To assess the performance of catalysts at industrial-relevant conditions, a home-customized flow cell based on $Cu_{97}Sn_3$ gas diffusion electrode was built for high current $CO_2RR$ (Supplementary Fig. 14). As shown in Fig. 4a, the polarization curve obtained with flow cell exhibits significantly reduced onset potential (~400 mV) compared with that in an H-cell, and achieves 10 mA cm$^{-2}$ at only −0.19 V vs. RHE. This onset potential of CO evolution on $Cu_{97}Sn_3$ reaches almost its theoretical potential of −0.11 V vs. RHE[41], which is the lowest among the state-of-the-art $CO_2$-to-CO catalysts, to the best of our knowledge (Supplementary Table 3). The low overpotential can be attributed to (i) enhanced $CO_2$ mass transport at the three-phase interfaces of catalyst, $CO_2$ gas, and KOH electrolyte; (ii) the catalytic promotion effect of hydroxide ions, which can lower the $CO_2$ activation energy barriers[42]. Fig. 4b displays the CO FEs and partial current densities plotted against the iR-corrected potentials. At a small overpotential of 30 mV, the $Cu_{97}Sn_3$ shows a high $FE_{CO}$ of 98% with $j_{CO}$ of 2.0 mA cm$^{-2}$. The $j_{CO}$ increases sharply with the applied potentials, reaching 100 mA cm$^{-2}$ at −0.45 V with a $FE_{CO}$ of 87%, and further exceeds 200 mA cm$^{-2}$ at −0.65 V with a $FE_{CO}$ of 67%. The overpotential of $Cu_{97}Sn_3$ at ~100 mA cm$^{-2}$ also outperforms most, if not all, the catalysts in recent reports.

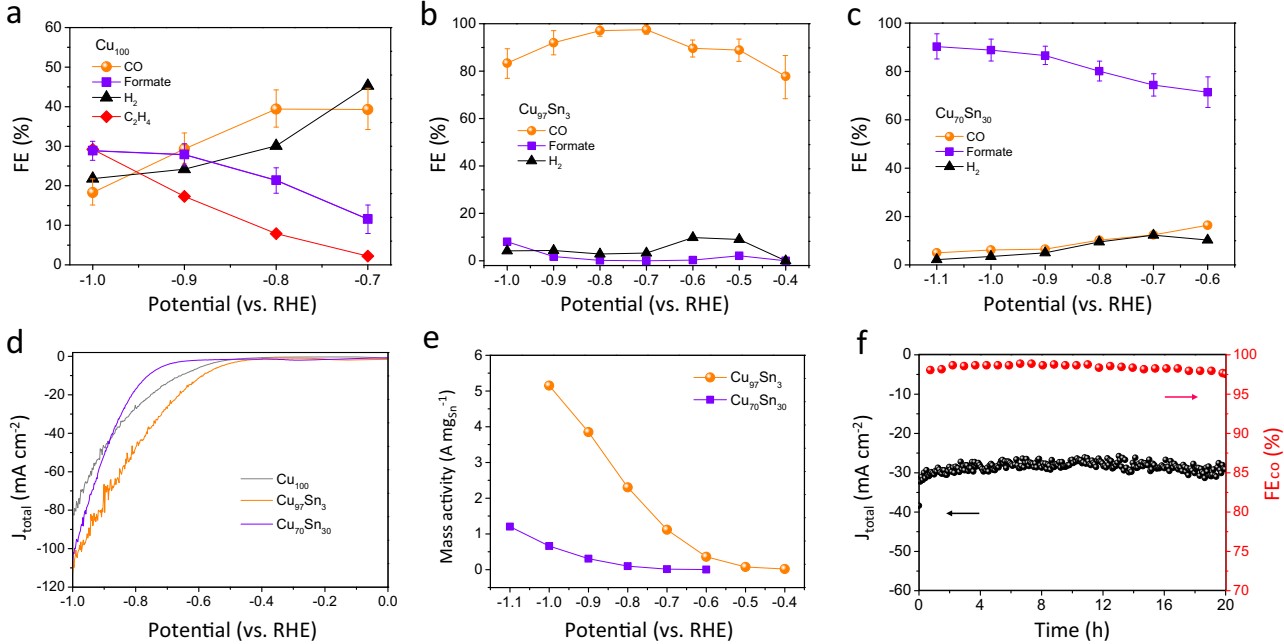

**Fig. 3 CO₂ electrolysis in an H-cell.** Potential dependence of Faradaic efficiencies for CO₂RR on $Cu_{100}$ (**a**), $Cu_{97}Sn_3$ (**b**), and $Cu_{70}Sn_{30}$ (**c**). **d** LSV curves at a scan rate of 10 mV s⁻¹. **e** Sn mass-normalized CO₂RR activity. **f** Stability test of $Cu_{97}Sn_3$ at −0.75 V vs. RHE.

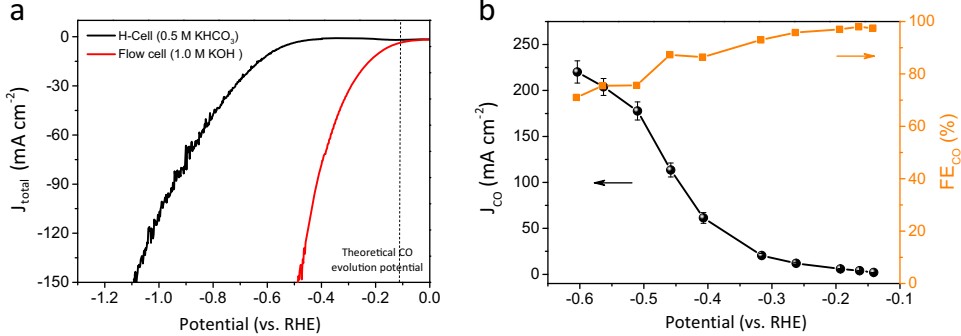

**Fig. 4 CO₂ electrolysis on $Cu_{97}Sn_3$ in a flow cell. a** Polarisation curves obtained at a scan rate of 10 mV s⁻¹ for the H-cell and the flow cell. **b** CO FEs and current densities at different potentials.

## Discussion

**Density functional theory (DFT) simulations**. To understand the different electrocatalytic behaviors of the catalysts, the thermodynamic reaction energetics on four different models including pure Cu, Cu–Sn surface alloy, Cu–Sn bulk alloy, and Cu–Sn core-shell bulk alloy were studied using DFT simulations (Fig. 5a). The corresponding reaction pathways involve $CO_2$–COOH*–CO*–CO (Fig. 5b), $CO_2$–COOH*–HCOOH and $CO_2$–OCHO*–HCOOH (Supplementary Figs. 15, 16), and $H^+$–H*–$H_2$ (Fig. 5c), which have been proposed for CO₂RR on Cu–Sn alloys[29]. According to the XRD results, we used a face-centered cubic (*fcc*) model for Cu, and this structure has the lowest formation energies among the different phases[43]. Here, we considered stepped facets in our models, which have been found to be generally more active for CO₂RR than flat terrace sites[29,35].

On the pure Cu surface, the potential limiting steps for CO and HCOOH formation are the desorption of CO* and COOH*→HCOOH (Fig. 5b and Supplementary Fig. 16a), with the corresponding overpotentials of 0.39 and 0.24 eV, respectively. Given the tiny overpotential of 0.05 eV for HER on pure Cu (Fig. 5c), its activity and selectivity for CO₂RR are poor.

Besides, the strong absorption of CO* on Cu surface (−0.09 eV) is important for the formation of $C_2H_4$. When Cu–Sn surface alloy forms, the potential limiting steps for the CO and HCOOH production are the same, i.e., the formation of COOH*, with the overpotentials of 0.29 V (Supplementary Fig. 16b). Considering the much lower free energy of CO* (0.10 eV) compared with HCOOH (0.38 eV), COOH* tends to be reduced to CO* instead of HCOOH from a thermodynamics point of view. Moreover, the overpotential for HER on Cu–Sn surface alloy increases to 0.1 eV (Fig. 5c). All these results indicate that the isolated Cu–Sn bonding can tune the selectivity for CO₂-to-CO conversion with suppressed HER. As for both Cu–Sn bulk alloys, the most significant changes are the increase of COOH* free energy up to 0.66 and 0.81 eV (Supplementary Fig. 16c, d), respectively. Given the much lower free energy of OCHO* compared with COOH*, the CO₂-to-HCOOH conversion would dominate the CO₂RR process on Cu–Sn bulk alloys. Meanwhile, due to the relatively large overpotential for HCOOH production on core-shell bulk alloy (0.53 eV, Fig. 5d), a much higher negative potential is required to achieve desired currents. Collectively, all DFT results are in line with the experimental observations that, compared

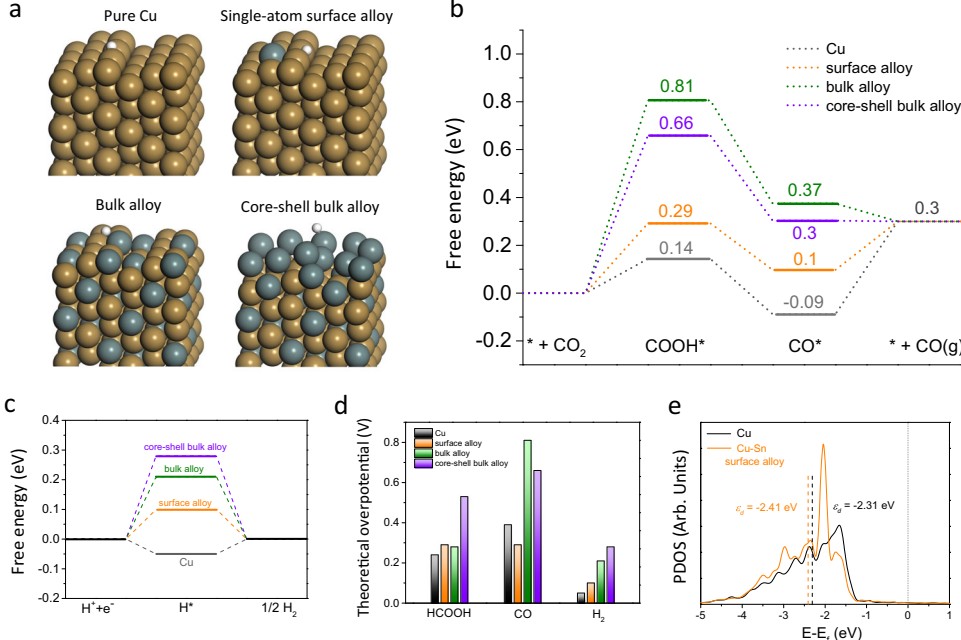

**Fig. 5 DFT simulations of CO₂RR on Cu–Sn atomic interfaces. a** The theoretical model structures on stepped facets of pure Cu, Cu–Sn surface alloy, bulk alloy and core-shell bulk alloy with adsorbed H*. The dark goldenrod and slate gray balls represent Cu and Sn atoms, respectively. The structures of the intermediates are shown in Supplementary Fig. 15. **b** The calculated free energy diagrams of CO₂-to-CO conversion. **c** The calculated free energy diagrams of HER. **d** The theoretical overpotentials of HCOOH, CO, and H₂ production on catalysts. **e** The projected density of states (PDOS) of $d$ orbitals of active Cu atoms at the step edges on pure Cu and Cu–Sn surface alloy. The dotted gray line indicates the Fermi level. The $d$-band centers ($\varepsilon_d$) are denoted by dashed lines.

with pure Cu surface, $Cu_{97}Sn_3$, and $Cu_{99}Sn_1$ surface alloys exhibit superior activity and selectivity for CO production while $Cu_{70}Sn_{30}$ bulk alloy shows excellent reactivity for formate generation at large overpotentials.

According to $d$-band theory, the behavior of the occupied $d$ orbital projected on the catalyst surface tightly correlates with the local electron transfer and surface chemisorption. In the context of chemisorption of molecules to a metal surface, a more positive $d$-band center manifests a stronger adsorption interaction between the metal sites and the adsorbates[44,45]. To explain the different electrocatalytic behaviors of our catalysts, we further calculated the projected density of states (PDOS) of $d$ orbitals and the $d$-band center ($\varepsilon_d$) of Cu atoms (Fig. 5e). The DFT calculations show an obvious left shift of PDOS of Cu $d$ orbitals in surface alloy compared to pure Cu, and the corresponding $d$-band center is thus farther to the Fermi level than that on pure Cu by 0.1 eV. As a result, the bonding of COOH*, H*, and OCHO* are all weakened on isolated Cu–Sn bonding to a certain extent, which increases the $U_L(CO_2)$–$U_L(H_2)$ for CO production (Supplementary Fig. 17). To sum up, the unique electrocatalytic behavior of isolated Cu–Sn atomic interfaces is attributed to the combination of geometric (different coordination environment) and electronic (change of $d$ orbitals) effects, which can explain the different CO₂RR reactivity of pure Cu, $Cu_{97}Sn_3$ surface alloy, and $Cu_{70}Sn_{30}$ core-shell bulk alloy.

In this work, pure Cu, single-atom surface alloys ($Cu_{97}Sn_3$ and $Cu_{99}Sn_1$), and core-shell bulk alloy ($Cu_{70}Sn_{30}$) were synthesized via a one-step sequential reduction process. Extensive structural characterizations revealed a high surface density (up to 8%), isolated Sn sites in $Cu_{97}Sn_3$, which is able to break the linear scaling relationship of Cu or Sn catalysts. As such, the observed distinct CO₂RR selectivity and orders-of-magnitude increase in Sn-mass-normalized activity on $Cu_{97}Sn_3$ relative to $Cu_{70}Sn_{30}$ is understood as a result of its unique surface structure and bonding

geometry. In an alkaline flow cell, we achieved a high $FE_{CO}$ of 98% at a tiny overpotential of 30 mV and a $j_{CO}$ of 100 mA cm$^{-2}$ at $-0.45$ V vs. RHE. In light of this superior performance, together with the high degree of control over the atomic structure, it is expected that the strategy could be applied for the design of various surface alloy electrocatalysts for CO₂RR, as well as a range of electrochemical energy conversion reactions such as oxygen reduction, nitrogen fixation, and beyond.

## Methods

**Synthesis of Cu-Sn nanoparticle alloys.** The Cu–Sn nanoparticles were synthesized via a simple one-step reduction method based on copper (II) chloride (CuCl₂), tin (II) chloride (SnCl₂), and sodium borohydride (NaBH₄). Since the Cu and Sn can be easily oxidized at ambient conditions, a concentrated strong reduction solution (5 M NaBH₄) was used to ensure the completed reduction. For the preparation of the $Cu_{100}$ sample, 2 mL CuCl₂ (300 mg) aqueous solution was rapidly added into 2 mL NaBH₄ solution under ice bath and aged for 0.5 h. The as-prepared black suspension was then washed with H₂O and acetone several times. After drying in vacuum at room temperature, the $Cu_{100}$ nanoparticles were obtained. For the synthesis of $Cu_{99}Sn_1$ and $Cu_{97}Sn_3$, 4.1 mg and 12.3 mg SnCl₂ was added into 2 mL CuCl₂ (300 mg) solution, respectively, before further react with NaBH₄. For the preparation of the $Cu_{70}Sn_{30}$ sample, 300 mg CuCl₂ and 170.2 mg SnCl₂ were added into 3.2 mL H₂O and then react with NaBH₄. After synthesis, all the samples were stored under vacuum at room temperature to avoid oxidation for further use. The ratio of Sn and Cu in alloys were calculated from the stoichiometry of the precursor metals in the chemical reduction process.

**Characterizations.** Scanning electron microscope (SEM) images were collected with a QUANTA 450. Transmission electron microscopy (TEM), high-resolution TEM (HRTEM), high-angle annular dark-field scanning TEM (HAADF-STEM) were carried out on JEOL JEM-ARM200f microscope at 200 kV. XRD was performed on a PANalytical X'Pert XRD system (45 kV, 40 mA, Cu Kα radiation). XPS results were recorded by Thermo ESCALAB250Xi. Cu LMM Auger peak was obtained by the region scan of 555–590 eV Binding Energy, which were then converted to the Kinetic Energy. UV–vis spectroscopy was carried out on Varian Cary 100 Scan double-beam UV/Vis spectrophotometer. XAFS measurement and data analysis: XAFS spectra at the Cu K-edge and Sn K-edge were collected at Beijing Synchrotron Radiation Facility and 12-BM-B at the Advanced Photon

Source in Argonne National Laboratory, respectively. The Cu and Sn K-edge XANES data were recorded in a transmission mode.

**Atom probe tomography (APT).** The specimen was prepared in a Zeiss Auriga scanning electron microscopy/focused ion beam (SEM/FIB) using a modified liftout technique. The particle suspension was initially made by sonicating their ethanol mixture for 15 min and was dispersed on a Si substrate. After drying out, the substrate was welded on a SEM stub and transferred into SEM for particle liftout. The particles are prone to agglomerate into clusters with different sizes, and we target the one with a size of ~500 nm in diameter to facilitate our liftout experiment. The particle was easily adhered to the in-built tungsten manipulator by electrostatics and placed onto a pretreated Mo post. Thereafter, the gas-injection-system was used to deposit a thick Pt capping layer in order to firmly consolidate the specimen. Following a step by step FIB annual milling procedure, the specimen was sharpened less than 100 nm at the apex and was ready for APT experiment.

The identity of the chemicals can be determined by measuring the time of flight of each type of atom from the sample and detector. The data visualization and analysis can be carried out in a commercial software Integrated Visualization and Analysis (IVAS) 3.8.4. In this work, a CAMECA local electrode atom probe (LEAP4000X Si) assisted with a UV laser system (35–5 nm wavelength) was used for the experiment. The experiments were conducted at 50 K temperature, and laser energy, detection rate and pulse frequency were set at 20 pJ, 0.5% and 200 kHz respectively.

**Electrochemical measurements.** $CO_2$ electrolysis in H-cells was performed in a gas-tight H-cell with two-compartments separated by a cation exchange membrane (Nafion®117). A Pt plate was used as the counter electrode, a saturated calomel electrode (SCE) was used as the reference electrode, and $CO_2$-saturated 0.5 M $KHCO_3$ was used as the electrolyte, respectively. To prepare the working electrodes, 10 mg of catalyst and 100 μL of 5% Nafion solution were introduced into 100 μL of water and 300 μL of ethanol solution, and sonicated for 1 h. A 6.25 μL of the catalyst ink was coated onto a carbon fiber paper substrate and dried in air, giving a catalyst loading of 0.5 mg cm$^{-2}$. Before testing, all the samples were electro-chemically reduced by applying a high negative potential from $-0.5$ to $-2$ V vs. RHE at 50 mV s$^{-1}$ for ten cycles. All LSV and potentiostatic data were corrected with an $iR$ compensation of 80%. All potentials were calculated with respect to the reversible hydrogen electrode (RHE) scale according to the Nernst equation ($E_{RHE} = E_{SCE} + 0.0591 \times pH + 0.241$ V, at 25 °C).

$CO_2$ electrolysis in flow cells at industrially relevant conditions was carried out in a home-customized flow cell. The windows for electrolysis were set to 1 cm × 1 cm. Each chamber has an inlet and an outlet for electrolyte, and a SCE reference electrode was placed in the catholyte chamber. The catalyst ink was prepared by mixing 20 mg of catalyst, 6 mL of ethanol, and 200 μL of a Nafion perfluorinated resin solution. Then, catalysts were air-brushed onto 1.5 × 1.5 cm$^{-2}$ 25 BC gas diffusion layer (Fuel Cell Store) electrodes with mass loading of 1.0 mg cm$^{-2}$, and used as the cathode. A 1.5 × 1.5 cm$^{-2}$ Ni foam was used as a counter electrode for oxygen evolution reaction. An anion exchange membrane (Dioxide Materials) was used to separate the cathode and anode chambers. 1 M KOH solution was used as the electrolytes. The catholyte and anolyte were cycled at a flow rate of 50 mL min$^{-1}$ by using a peristaltic pump. The gas inlet and outlet on the cathode side were linked to a $CO_2$ gas-flow meter (30 sccm) and a GC, respectively. The applied potentials were converted to the RHE scale with $iR$ correction ($E_{RHE} = E_{SCE} + 0.0591 \times pH + 0.241$ V $+ iR$, at 25 °C)[46].

## Data availability
The data that support the findings of this study are available within the paper and its Supplementary Information File or are available from the corresponding authors upon reasonable request. Source data are provided with this paper.

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

## Acknowledgements

This work was supported by the Australian Research Council (FT170100224, DP160101713). The authors are thankful to Dr. Xianjue Chen for his help in conducting the TEM tests. The authors are thankful to BSRF (Beijing Synchrotron Radiation Facility), 12-BM-B at the Advanced Photon Source (APS) in Argonne National Laboratory, and UNSW Mark Wainwright Analytical Center for providing access to their XAFS, XRD, SEM, XPS, Raman, NMR, and facilities. The authors acknowledge the technical and scientific support of the Microscopy Australia node at the University of Sydney (Sydney Microscopy & Microanalysis). This research was also undertaken with the assistance of resources provided by the Pawsey and the National Computing Infrastructure (NCI) facility at the Australian National University; allocated through both the National Computational Merit Allocation Scheme supported by the Australian Government and the Australian Research Council grant LE190100021 (Sustaining and strengthening merit-based access at NCI, 2019-2021). This project has received funding from the European Union's Horizon 2020 research and innovation program under the Marie Sklodovska-Curie grant agreement No. 891545-ADBCRZB.

## Author contributions

C.Z. conceived and directed the project. W.R. designed the experiments, analyzed the results and wrote the first draft of the paper. X.T. and S.S. performed the DFT calculation. J.Q., S.R., and J.C. carried out the APT measurements. S.L., J.L., X.L., and K.W. undertook the XAFS characterization. All authors contribute to the writing of the paper.

## Competing interests

The authors declare no competing interests.
