## [Peer Review File · Nature Communications]

REVIEWER COMMENTS

Reviewer #1 (Remarks to the Author):

This paper presents a new 'single-atom-surface' Cu₉₇Sn₃ nanoparticle system that exhibits superior electrocatalytic performance for the CO₂ reduction reaction. The authors attributed the excellent electrocatalytic performance to the interface between the surface single Sn atoms and the Cu matrix. They proposed that the 'atomic interface' engineering can be a useful strategy for designing high-performance nanoparticles for CO₂ reduction reaction and other electrochemical conversion reactions. The concept proposed in this study is interesting. However, I am not convinced that it is supported by enough experimental data. The key experimental results showing the 'atomic interface' were obtained by STEM/EDX and atom probe tomography (APT). The STEM/EDX mapping (Fig. 1b) shows that the Cu₉₇Sn₃ nanoparticles are homogeneously alloyed (as illustrated by Fig. 1a). However, the APT data of Cu₉₇Sn₃ (Fig.2) shows an evident Sn surface segregation. Thus, I am not sure if the authors intend to claim that Cu₉₇Sn₃ is core-shell nanoparticles or alloyed nanoparticles. Can the authors clarify this point? Also, the APT analyses of nanoparticles are rather challenging due to difficulties with APT sample preparation and data interpretation. In the case of core-shell nanoparticles, the sample preparation is even more difficult. Because to identify the surface of the nanoparticles accurately, each particle must be separated from each other to avoid trajectory aberrations. The authors only showed the APT data of a single nanoparticle. Can the authors provide the entire APT data in the supplementary data? Without looking at the entire data, it is difficult to evaluate the elemental distribution on the nanoparticles' surfaces. Additionally, can the authors provide the mass spectra of the whole data? The Sn signal is rather low, and it is unclear if it is Sn or signals of impurities or surfactants introduced by sample preparation. Another concern is that the XRD data shows that the Cu₉₇Sn₃ nanoparticles are heavily oxidized. Such surface oxides can be clearly revealed by APT, but the authors did not report this point. Also, how does the oxide layer contribute to the performance of the CO₂ reduction reactions? Essentially, the electrocatalytic experiments were performed on the oxidized 'single-atom-surface' Cu₉₇Sn₃ nanoparticles.

Reviewer #2 (Remarks to the Author):

The manuscript "Isolated copper-tin atomic interfaces tuning electrocatalytic CO₂ conversion" by Zhao and co-workers describes a novel way of tuning the selectivity in CO₂RR by careful design of CuSn single-atom surface alloys. Experimentally, isolated Sn atoms are shown to be preferentially located on the surface of the Cu nanoparticles. The authors demonstrate how Cu₉₇Sn₃ and Cu₉₉Sn₁ can convert CO₂ to CO with extremely high efficiency and very low overpotential (only 30 mV in alkaline flow cell) and current density of 100 mA cm⁻² at -0.45 V vs RHE, while Cu₇₀Sn₃₀ bulk alloy preferentially forms formate. The results are elegantly compared to pure Cu and bulk alloy systems and the unique performance is explained by DFT calculations to originate from the electronic effects of the local coordination environment.

The paper is suitable for Nature Communications and I recommend its publication after addressing a few minor questions.

1. What is the reason for performing gas-phase calculations and then applying correction terms on the adsorbed species for the solvation effects rather than using the PCM method?
2. Have the transition states for the corresponding CO and HCOOH products being located and if so, do they confirm the trend in Figure 5b?
3. Throughout the text on pages 7-9 Cu₇₀Sn₃₀ is written as Cu₉₇Sn₃ as a result of copy pasting. Please fix that inconsistency.
4. What is the expected limit in alloy compositions between Cu₉₇Sn₃ and Cu₇₀Sn₃₀ which will still produce preferentially CO instead of formate? At which composition the formation of both products will occur with similar efficiency?

5. Is there another alloy composition that can outperform or do similar as Cu₉₇Sn₃? If not, what could be the next step for design improvements of these alloy materials?

Reviewer #3 (Remarks to the Author):

Herein, the authors have investigated the CO₂ electroreduction in Cu-Sn particles with dispersed or atomic single sites of Sn particles on a Cu matrix, prepared by convectional chemical reduction in borohydride media. They have characterized the nanoparticles and assessed the dispersion degree of Sn by combining different characterization techniques: STEM/EDX mapping and HAADF-STEM and Atom Probe microscopy. They also provided XRD, XPS and XAFS measurements, among other techniques. The experimental data that they have presented shows clear evidence of both the presence and dispersion of Sn atoms on the Cu matrix.

Promising selectivity and activities values towards the production of either CO or HCOOH at different applied potentials were obtained in an alkaline flow cell, for further industrial application. Tests of catalyst stability were also conducted showing that the prepared bimetallic nanoparticles have a good robustness with time.

Finally, the singular electrocatalytic performance of atomic dispersed Sn on Cu nanoparticles, was rationalized by means of DFT modeling, which strongly suggests that the bond between Cu-isolated Sn atoms introduces both geometrical and electronic effects which are the main responsible of the electrocatalytic improvement. Importantly, experimental results have been compared with modelled stepped Cu surfaces, which approach more to real conditions. Because of these reasons, I believe that this article is suitable for Nature Communications. I also believe that the paper shed some lights on tuning the electrocatalytic properties of Cu by tailoring the surface active site with the incorporation of single atomic sites of other elements, in this case Sn. However, I have some questions and comments that I think the author should address:

1) How do you know the real ratio of Sn and Cu in either Cu₇₀Sn₃₀ or Cu₉₇Sn₃ particles. Was the Cu/Sn ratio calculated from the stoichiometry of the precursor metals in the chemical reduction process, or did you carry out chemical analysis such as ICP analysis (Inductively coupled plasma mass spectrometry) or via other techniques. Could you clarify this in the new version?

2) Do you have an idea of the degree of Sn isolated atoms on surface or in the sub-monolayers? I think the obtained data from the characterization techniques provides information of element distribution not at the surface level, i.e., it extends to different submonolayers. And, in relation to this question, are only single atoms of Sn in the surface-layer responsible of the electrocatalytic enhancement? Can subsurface Sn influence the CO₂ reduction in these bi-metallic nanoparticles in a similar or different way?

3) Fig 3a,b and c: Which is the uncertainty or error bar of the different efficiencies obtained at different applied potentials. I think this is important to show.

4) Fig S9. I think capacitance measurements are not particularly accurate to assess the electroactive surface area. I guess you assumed that the respective flat polycrystalline surfaces, i.e., with roughness factor 1, had similar capacitance constant value, and this is not necessarily true. But capacitance measurements are one of the most widely employed methodologies in literature to get an idea about the order of magnitude of the ECSA. I suggest the authors to add just a few references in which other researchers have used capacitance measurements for ECSA estimation in bi or multimetallic nanoparticles. In addition, I also suggest the authors to include the whole blank CVs in the S.I (i.e. between HER onset and onset of oxide formation) of Cu-Sn nanoparticles and Cu(100). Then, they should indicate which range of potential from the blank CV was selected to perform capacitance measurements. Sometimes, capacitance measurements could be conditioned by the potential limits.

5) Was the IR compensation the same in all the employed nanoparticulated catalyst surfaces?

Response Letter

Nature Communications (NCOMMS-20-39314A)

Isolated copper-tin atomic interfaces tuning electrocatalytic CO₂ conversion

Wenhao Ren,^{1,†} Xin Tan,^{2,†} Jiangtao Qu,^{3,4} Sesi Li,⁵ Jiantao Li,⁶ Xin Liu,⁵ Simon P. Ringer,³ Julie M. Cairney,^{3,4} Kaixue Wang,⁵ Sean C. Smith² and Chuan Zhao^{1,*}

Response to Reviewer #1:

General Comment: This paper presents a new 'single-atom-surface' Cu₉₇Sn₃ nanoparticle system that exhibits superior electrocatalytic performance for the CO₂ reduction reaction. The authors attributed the excellent electrocatalytic performance to the interface between the surface single Sn atoms and the Cu matrix. They proposed that the 'atomic interface' engineering can be a useful strategy for designing high-performance nanoparticles for CO₂ reduction reaction and other electrochemical conversion reactions.

Response to General Comment: We are thankful for reviewer #1's comments and suggestions on our manuscript. We welcome the opportunity to address and clarify the issues raised in the reviewer's report, and believe that the additional revisions will strengthen our manuscript. The point-by-point responses are as follows.

Comment-1. The concept proposed in this study is interesting. However, I am not convinced that it is supported by enough experimental data. The key experimental results showing the 'atomic interface' were obtained by STEM/EDX and atom probe tomography (APT). The STEM/EDX mapping (Fig. 1b) shows that the Cu₉₇Sn₃ nanoparticles are homogeneously alloyed (as illustrated by Fig. 1a). However, the APT data of Cu₉₇Sn₃ (Fig.2) shows an evident Sn surface segregation. Thus, I am not sure if the authors intend to claim that Cu₉₇Sn₃ is core-shell nanoparticles or alloyed nanoparticles. Can the authors clarify this point?

Response: Thanks. The Cu₉₇Sn₃ obtained in our study is a single-atom surface alloy, where the isolated Sn atoms are mainly anchored on the Cu host surface (Figure 1a). However, it is not a complete core-shell structure because the content of Sn is very low and cannot fully cover the surface of Cu nanoparticles. To make this conclusion clear, the overlapped EDX mapping together with line profile analysis of Cu₉₇Sn₃ were carried out (see below & New Figure S1), which clearly show the surface segregation of Sn and is consistent with the APT results.

Besides, as illustrated in Figure 1a, the reduction of Cu²⁺ takes place first during the synthesis, followed by the deposition of Sn on Cu surface (or near the surface). This sequential reduction process can be attributed to the more negative standard electrode potential of Sn²⁺ relative to Cu²⁺ ($\text{Sn}^{2+} + 2\text{e}^- \rightleftharpoons$

Sn, $E^0 = -0.13$ V vs. SHE; $\text{Cu}^{2+} + 2\text{e}^- \rightleftharpoons \text{Cu}$, $E^0 = 0.34$ V). As a result, the Sn is preferentially deposited onto the outer shell of the alloy nanoparticles for both $\text{Cu}_{97}\text{Sn}_3$ and $\text{Cu}_{70}\text{Sn}_{30}$, rather than a homogeneous bulk alloy.

Supplementary Figure 1 | Structural characterization of $\text{Cu}_{97}\text{Sn}_3$. (a) The HAADF-STEM image. (b) The corresponding overlapped EDX mapping of Cu (green) and Sn (red). (c) Line profile analysis derived from a.

Comment-2. Also, the APT analyses of nanoparticles are rather challenging due to difficulties with APT sample preparation and data interpretation. In the case of core-shell nanoparticles, the sample preparation is even more difficult. Because to identify the surface of the nanoparticles accurately, each particle must be separated from each other to avoid trajectory aberrations. The authors only showed the APT data of a single nanoparticle. Can the authors provide the entire APT data in the supplementary data? Without looking at the entire data, it is difficult to evaluate the elemental distribution on the nanoparticles' surfaces. Additionally, can the authors provide the mass spectra of the whole data? The Sn signal is rather low, and it is unclear if it is Sn or signals of impurities or surfactants introduced by sample preparation.

Response: According to the reviewer's suggestion, we have added the all-atom mixed tomography and the full-range mass to charge spectrum in the supplementary (Figure S3), and the following discussions have been added to the revised manuscript:

The whole APT experiment includes three stages. During Stage 1, the $\text{Cu}_{97}\text{Sn}_3$ alloy was successfully evaporated by applying electric pulses at the voltage of 1,600 V, which is a common voltage observed

for alloy samples (Supplementary Fig. 3a). After the stable evaporation, we obtained the tomography of a single nanoparticle (Supplementary Fig. 3b). Then, the voltage started to increase in Stage 2 because of the detection of the C (Pt) signals (Supplementary Fig. 3c). The C (Pt) comes from our specimen preparation process, during which we incorporated the nanoparticles into a layer of electron-beam-assisted deposition of $(C_5H_4)CH_3Pt(CH_3)_3$. This deposition can fill the interspace of each nanoparticles and make the tip compacted. However, the carbon needs higher energy to be evaporated so that the voltage gradually increases.¹² As the experiment continued to run, the carbon layer became thicker and the voltage even increased to above 5,000 V after Stage 3 (Supplementary Fig. 3d). In this case, the APT experiment would automatically end, and we cannot obtain more tomography of nanoparticles.

Supplementary Figure 3 | Atom probe tomography analysis of Cu₉₇Sn₃ specimen. (a) Voltage curves. 3D tomography at the end of Stage 1 (b), Stage 2 (c), and Stage 3 (d). (e) The full-range mass to charge spectrum.

The full-range mass-to-charge spectrum is shown in Supplementary Fig. 3e. Our specimen preparation method uses a gas-injection-system as adhesive to bond the nanoparticle onto the Mo post, followed by the (C₅H₄)CH₃Pt(CH₃)₃ deposition and a Ga source FIB annular milling.¹³ As a result, C, Pt, H, and Ga are the primary contamination introduced to the sample. Note that all the elements detected in the spectrum (H, C, Sn, Cu, Ga and Pt) have no overlapping owing to their different mass-to-charge-state ratio. Besides, the overall Sn content is more than 1% in the specimen and the surface content can reach up to ~8%, which is far beyond the detection limit of APT technique (ppm elemental detection capacity).¹⁴ Based on the above analysis, we can firmly claim that the signal peak at ~60 (Da) originating from Sn only, rather than impurities or noise.

Furthermore, to understand the surface segregation of Sn atoms from the thermodynamical point of view. New density function theory (DFT) simulations have been carried out to investigate the possible position of Sn atom in single-atom Cu-Sn alloy, i.e. if the Sn prefers to sit on the surface or in the bulk. We calculated the relative substitution energy, which corresponds to the difference between the energy of substitution of a Cu atom in an inner layer and the energy of substitution of a Cu atom at the surface, as shown in Figure S17a. The results suggest that the substitution energy barrier of Sn atoms significantly increases to ~1.5 eV for 2L(b) and 3L(d) position. Therefore, the Sn atoms prefer to anchor on the surface of Cu nanoparticles rather than in bulk from an energetics perspective.

Supplementary Figure 17 | (a) The relative substitution energy of a Cu atom by an Sn atom at different positions. All are given in eV.

To sum up, two important conclusions can be drawn from the APT technique which is complementary to STEM/EDX mapping: *i*) the atomic isolation of Sn atoms on Cu nanoparticles from a three-dimensional perspective (Supplementary Movie 01); *ii*) the quantitative analysis of Sn content on Cu host based on the proxigram profile, which suggests a high Sn content up to ~8% on the surface (Fig. 2b-e). Therefore, based on the STEM/EDX mapping, APT analysis, DFT calculation, and the higher standard electrode potential of Cu²⁺ vs Sn²⁺ (Sn²⁺ + 2e⁻ ⇌ Sn, E⁰ = -0.13 V vs. SHE; Cu²⁺ + 2e⁻ ⇌ Cu, E⁰ = 0.34 V), we conclude that the Cu₉₇Sn₃ is a single-atom surface alloy, where the isolated Sn atoms are mainly anchored on the Cu host surface.

[12] Thuvander, M. *et al.* Quantitative atom probe analysis of carbides. *Ultramicroscopy* **111**, 604-608 (2011).

[13] Felfer, P. *et al.* New approaches to nanoparticle sample fabrication for atom probe tomography. *Ultramicroscopy* **159**, 413-419 (2015).

[14] Gault, B., Moody, M. P., Cairney, J. M. & Ringer, S. P. *Atom Probe Microscopy*. Vol. 160

(Springer Science & Business Media, 2012).

Comment-3. Another concern is that the XRD data shows that the Cu₉₇Sn₃ nanoparticles are heavily oxidized. Such surface oxides can be clearly revealed by APT, but the authors did not report this point. Also, how does the oxide layer contribute to the performance of the CO₂ reduction reactions? Essentially, the electrocatalytic experiments were performed on the oxidized 'single-atom-surface' Cu₉₇Sn₃ nanoparticles.

Response: We agree with the reviewer about the presence of the oxides, as suggested by XRD and XPS data. However, our APT data does not show significant O peak in the mass spectrum (see Figure S3e). This is possibly because during the APT experiment, focused ion beam (FIB) annular milling was employed to remove the surface nanoparticles and finally the tip formation stops at the center region of the cluster. Thus, the nanoparticles in the center of the cluster are protected by the surrounded surface particles and consequently were not oxidized. Therefore, the mass spectrum didn't show an obvious O signal. Please see the detailed sample preparation process in Figure R1.

Figure R1 | Specimen preparation of atom probe tomography. APT requires a needle-like specimen with a diameter less than 100 nm, to retain sufficient local electrical field around the needle apex for atoms field evaporation. The specimen was prepared in a Zeiss Auriga scanning electron microscopy/focused ion beam (SEM/FIB) in terms of a modified lift-out technique. The particle suspension was initially made by sonicating their ethanol mixture for 15 mins and was dispersed on a Si substrate. After drying out, the substrate was welded on an SEM stub and transferred into SEM for particle liftout. We then target on the nanoparticle cluster with a size of ~500 nm in diameter to facilitate our liftout experiment (Figure R1a,b). The particle was easily adhered to the in-built tungsten manipulator by electrostatics and placed onto a pre-treated Mo post (Figure R1c,d). Thereafter, the gas-injection-system (gas) was used to deposit a thick (C₅H₄)CH₃Pt(CH₃)₃ filler in order to firmly consolidate the specimen (Figure R1e). Following a step-by-step FIB annual milling procedure to remove surface particles, the specimen was sharpened less than 100 nm and finally the tip formation stops at the center region, and was ready for the APT experiment (Figure R1f).

We are aware that the pristine catalysts were inevitably oxidized when exposed to the air during electrode preparation. Depending on the exposure time, the oxidation degree is different. On the other hand, negative potentials (from -0.4 V to -1.1 V) were applied to the electrode upon CO₂RR

measurements. With different applied potentials and time, the oxidized CuSn catalysts could be electrochemically reduced to different levels. In our study, to exclude the influence of surface oxides, cyclic voltammetry (CV) pretreatment was conducted at relatively negative potentials from -0.5 to -2 V vs. RHE for 10 cycles prior to CO₂RR (Supplementary Fig. 7). As confirmed by the *ex-situ* Sn 3d XPS and Cu LMM Auger spectra, most of the oxides on Cu₉₇Sn₃ were reduced after the CV pretreatment (Supplementary Fig. 8). This result is consistent with their standard electrode potentials ($\text{Cu}^+ + \text{e}^- \rightleftharpoons \text{Cu}$, $E^0 = +0.52$ vs SHE and $\text{Sn}^{2+} + 2\text{e}^- \rightleftharpoons \text{Sn}$, $E^0 = -0.13$ vs SHE).

Supplementary Figure 7 | CV pretreatment. (a) Pristine LSV curves of Cu₉₇Sn₃₀. (b-d) CV pretreatment from -0.5 to -2.0 V vs. RHE for 10 cycles at 50 mV s⁻¹.

Supplementary Figure 8 | Structural analysis of $\text{Cu}_{97}\text{Sn}_3$ before and after CV treatment. (a) XPS of Sn 3d spectra. (b) Cu LMM Auger spectra. *Ex-situ* Sn 3d XPS and Cu LMM Auger spectra on $\text{Sn}_{97}\text{Sn}_3$ confirm the reduction of Sn^{2+} to Sn^0 and Cu^+ to Cu^0 after the CV pretreatment. The residual oxides can be attributed to the inevitable oxidation during sample delivery and test. It is expected that most of the Cu and Sn species are reduced to the metallic state after the CV treatment, according to their standard reduction potential ($\text{Cu}^+ + \text{e}^- \rightleftharpoons \text{Cu}$, $E^0 = +0.52$ vs. SHE and $\text{Sn}^{2+} + 2\text{e}^- \rightleftharpoons \text{Sn}$, $E^0 = -0.13$ vs. SHE).^{6, 8, 12}

To further investigate the influence of the surface oxide layer, the as-synthesized $\text{Cu}_{97}\text{Sn}_3$ sample is exposed to the air and aged for 1 week before CO_2RR testing. As shown in XRD analysis, the $\text{Cu}_{97}\text{Sn}_3$ catalyst is further oxidized after continuous exposure to the air. While, after the CV pretreatment, the aged $\text{Cu}_{97}\text{Sn}_3$ show very close LSV curves and similar FE_{CO} compared with pristine $\text{Cu}_{97}\text{Sn}_3$ for CO_2 electrolysis, indicative of the similar CO_2RR reactivity and the negligible effect of surface oxides after CV pretreatment, which is consistent with our structural analysis (Supplementary Fig. 8).

New Supplementary Figure 9 | CO_2RR assessment of $\text{Cu}_{97}\text{Sn}_3$ after exposure to the air for 1 week. (a) XRD analysis. (b) LSV comparison. (c) FE_{CO} from -0.4 to -1.0 V vs RHE.

Response to Reviewer #2:

General Comment: The manuscript "Isolated copper-tin atomic interfaces tuning electrocatalytic CO₂ conversion" by Zhao and co-workers describes a novel way of tuning the selectivity in CO₂RR by careful design of CuSn single-atom surface alloys. Experimentally, isolated Sn atoms are shown to be preferentially located on the surface of the Cu nanoparticles. The authors demonstrate how Cu₉₇Sn₃ and Cu₉₉Sn₁ can convert CO₂ to CO with extremely high efficiency and very low overpotential (only 30 mV in alkaline flow cell) and current density of 100 mA cm⁻² at -0.45 V vs RHE, while Cu₇₀Sn₃₀ bulk alloy preferentially forms formate. The results are elegantly compared to pure Cu and bulk alloy systems and the unique performance is explained by DFT calculations to originate from the electronic effects of the local coordination environment. The paper is suitable for Nature Communications and I recommend its publication after addressing a few minor questions.

Response: We very much appreciate reviewer #2 for the time and high evaluation of our work. We have carried out new experiments and revised our work accordingly, our point-by-point responses are as follows.

Comment-1. What is the reason for performing gas-phase calculations and then applying correction terms on the adsorbed species for the solvation effects rather than using the PCM method?

Response: Thanks. This method has been widely used for DFT computation of CO₂RR on metal surfaces, and works very well (e.g. *Nat. Catal.* **2019**, 2, 55-61, *Energy. Environ. Sci.* **2010**, 3, 1311-1315, *PNAS* **2020**, 117, 1330-1338). On the other hand, the calculated solvation energies using PCM method show significant model dependence, and the accurate computation of solvation energy remains a grand challenge. Therefore, we choose performing gas-phase calculations and then applying correction terms on the adsorbed species for the solvation effects in our DFT calculations.

Comment-2. Have the transition states for the corresponding CO and HCOOH products being located and if so, do they confirm the trend in Figure 5b?

Response: For CO₂RR on Cu-Sn alloy surface, we considered different reaction pathways:
Formate:

Carbon monoxide:

These electrochemical reactions involve the proton-electron transfer process, which is a challenge for DFT computation. On the other hand, the widely used "Computational Hydrogen Electrode" (CHE) model, which is developed by Norskov (*Energy. Environ. Sci.* 2010, 3, 1311-1315, *J. Phys. Chem. B* 2004, 108, 17886) and relies on thermodynamic predictions, such as intermediate adsorptions and Gibbs free energies, fits well with CO₂ reduction reaction on the metal surface. Therefore, in our DFT calculations, we have used CHE model and haven't considered the transition states for CO and HCOOH products.

Comment-3. Throughout the text on pages 7-9 $\text{Cu}_{70}\text{Sn}_{30}$ is written as $\text{Cu}_{97}\text{Sn}_3$ as a result of copy pasting. Please fix that inconsistency.

Response: We thank the reviewer for the careful reminder. The spelling in the whole manuscript has been carefully checked and corrected.

Comment-4. What is the expected limit in alloy compositions between $\text{Cu}_{97}\text{Sn}_3$ and $\text{Cu}_{70}\text{Sn}_{30}$ which will still produce preferentially CO instead of formate? At which composition the formation of both products will occur with similar efficiency?

Response: According to the reviewer's suggestion, we have further synthesized $\text{Cu}_{95}\text{Sn}_5$, $\text{Cu}_{90}\text{Sn}_{10}$, and $\text{Cu}_{80}\text{Sn}_{20}$ catalysts for CO_2RR evaluation, and the following discussions have been added to the revised manuscript:

“As shown in Supplementary Fig. 11, the $\text{Cu}_{95}\text{Sn}_5$ exhibits CO dominated reactions from -0.6 to -1.0 V vs RHE, and the corresponding FE_{CO} reaches 98% at -0.7 V. As for $\text{Cu}_{90}\text{Sn}_{10}$, the FE_{CO} outperforms $\text{FE}_{\text{Formate}}$ at low potentials from -0.6 to -0.7 V, and then the production of CO and formate becomes similar at -0.8 V with FE of both products at ~45%. With the further increase of overpotential, the $\text{Cu}_{90}\text{Sn}_{10}$ produces preferentially formate instead of CO. As for $\text{Cu}_{80}\text{Sn}_{20}$, the $\text{FE}_{\text{Formate}}$ exceeds the FE_{CO} in the whole potential range and more approaches to the performance of $\text{Cu}_{70}\text{Sn}_{30}$.”

New Supplementary Figure 11 | CO_2RR measurements on $\text{Cu}_{95}\text{Sn}_5$ (a), $\text{Cu}_{90}\text{Sn}_{10}$ (b), and $\text{Cu}_{80}\text{Sn}_{20}$ (c).

Comment-5. Is there another alloy composition that can outperform or do similar as $\text{Cu}_{97}\text{Sn}_3$? If not, what could be the next step for design improvements of these alloy materials?

Response: The $\text{Cu}_{97}\text{Sn}_3$ generally exhibits the best CO_2 -to-CO electrocatalytic properties in our Cu-Sn alloy system. Nonetheless, at certain potentials (e.g. -0.7 V), the $\text{Cu}_{95}\text{Sn}_5$ shows similar CO_2RR performance compared with $\text{Cu}_{97}\text{Sn}_3$. We believe there will be two possible steps for the design improvements of these alloy materials. First, both the formate and CO are valuable products, and more

importantly, they are naturally separated. Therefore, it is possible to produce both products with one catalyst, and even produces a controlled ratio between CO and formate with the designed composition of Cu-Sn alloys. Second, other alloy compositions (e.g. Cu-Pd, Cu-In, Cu-Al, Cu-Ag, etc.) can be potentially designed and used for CO₂RR applications. Note that this single-atom surface alloy strategy only uses a small amount of foreign atoms, which also makes alloying with precious metal more practical.

Response to Reviewer #3:

General Comment: Herein, the authors have investigated the CO₂ electroreduction in Cu-Sn particles with dispersed or atomic single sites of Sn particles on a Cu matrix, prepared by convectional chemical reduction in borohydride media. They have characterized the nanoparticles and assessed the dispersion degree of Sn by combining different characterization techniques: STEM/EDX mapping and HAADF-STEM and Atom Probe microscopy. They also provided XRD, XPS and XAFS measurements, among other techniques. The experimental data that they have presented shows clear evidence of both the presence and dispersion of Sn atoms on the Cu matrix. Promising selectivity and activities values towards the production of either CO or HCOOH at different applied potentials were obtained in and alkaline flow cell, for further industrial application. Tests of catalyst stability were also conducted showing that the prepared bimetallic nanoparticles have a good robustness with time. Finally, the singular electrocatalytic performance of atomic dispersed Sn on Cu nanoparticles, was rationalized by means DFT modeling, which strongly suggests that the bond between Cu- isolated Sn atoms introduces both geometrical and electronic effects which are the main responsible of the electrocatalytic improvement. Importantly, experimental results have been compared with modelled stepped Cu surfaces, which approach more to real conditions. Because of these reasons, I believe that this article is suitable for Nature Communications. I also believe that the paper shed some lights on tuning the electrocatalytic properties of Cu by tailoring the surface active site with the incorporation of single atomic sites of other elements, in this case Sn. However, I have some questions and comments that I think the author should address:

Response: We very much appreciate reviewer #3 for the time and high evaluation of our work. After revision, we believe that the new insights will strengthen the revised manuscript, and our point-by-point responses are as follows.

Comment-1. 1) How do you know the real ratio of Sn and Cu in either Cu₇₀Sn₃₀ or Cu₉₇Sn₃ particles. Was the Cu/Sn ratio calculated from the stoichiometry of the precursor metals in the chemical reduction process, or did you carry out chemical analysis such as ICP analysis (Inductively coupled plasma mass spectrometry) or via other techniques. Could you clarify this in the new version?

Response: Thanks, the ratio of Sn and Cu in alloys were calculated from the stoichiometry of the precursor metals in the chemical reduction process. Because we use a very strong and concentrated reductant (5 M NaBH₄) for the synthesis, and believe most of the Cu and Sn species can be reduced. To reveal the real ratio of Sn and Cu in our samples, inductively coupled plasma optical emission spectroscopy (ICP-OES) analysis was also carried out for Cu₉₇Sn₃ and Cu₇₀Sn₃₀ samples (New Table S1). The following clarification has been added to the revised manuscript:

Table S1. Inductively coupled plasma optical emission spectroscopy (ICP-OES) analysis. The influence of oxygen is excluded.

Samples	Cu (at%)	Sn (at%)
Cu ₉₇ Sn ₃	97.3	2.7
Cu ₇₀ Sn ₃₀	72.8	27.2

“The ratio of Sn and Cu in alloys were calculated from the stoichiometry of the precursor metals in the

chemical reduction process.” (Experimental section)

Comment-2. Do you have an idea of the degree of Sn isolated atoms on surface or in the sub-monolayers? I think the obtained data from the characterization techniques provides information of element distribution not at the surface level, i.e., it extends to different submonolayers. And, in relation to this question, are only single atoms of Sn in the surface-layer responsible of the electrocatalytic enhancement? Can subsurface Sn influence the CO₂ reduction in these bi-metallic nanoparticles in a similar or different way?

Response: We agree with the reviewer that the Sn atoms exist not only on the surface but possibly in subsurface layers. In order to investigate the possible position of Sn atom in single-atom Cu-Sn alloy, i.e. if the Sn prefers to sit on surface or subsurface layers, we first calculated the relative substitution energy, which corresponds to the difference between the energy of substitution of a Cu atom in an inner layer and the energy of substitution of a Cu atom at the surface, as shown in Figure S17a. The results suggest that the Sn atom prefers to sit on the surface rather than in bulk from an energetics perspective.

To investigate the effects of Sn atoms from different layers, we also calculated the free energy diagrams of CO₂-to-CO conversion for Cu-Sn single-atom alloys with different Sn positions including 1L(a), 2L(b), 2L(c) and 3L(d) (Figure S17b). For Cu-Sn single-atom alloy with Sn at the surface, the overpotential of CO₂ to-CO conversion is very small (0.29 eV). If the Sn atom locates in the second or the third layer, the electrocatalytic performance also displays a certain degree of enhancement compared with pure Cu, however, the overpotentials of CO₂-to-CO conversion increase significantly to 0.34~0.43 V compared with at the surface. These results indicate that the CO₂RR activities of single-atom Cu-Sn alloys are dependent on Sn atom position, and the single-atom Cu-Sn alloy with Sn at the topmost surface exhibits the highest CO₂RR activity.

New Supplementary Figure 17 | DFT simulations of CO₂RR on Cu-Sn alloys with Sn atom in different layers. (a) The relative substitution energy of a Cu atom by an Sn atom at different positions. All are given in eV. (b) The calculated free energy diagrams of CO₂-to-CO conversion for Cu-Sn

single-atom alloys with different Sn position.

Comment-3. Fig 3a,b and c: Which is the uncertainty or error bar of the different efficiencies obtained at different applied potentials. I think this is important to show.

Response: Agreed. The error bar for the major products has been added to the revised manuscript to indicate the uncertainty of the corresponding Faradic efficiencies. Each error bar was the standard deviation determined based on tests of three individual electrodes.

Fig. 3 CO₂ electrolysis in an H-cell. Potential dependence of Faradaic efficiencies for CO₂RR on Cu₁₀₀ (a), Cu₉₇Sn₃ (b), and Cu₇₀Sn₃₀ (c). (d) LSV curves at a scan rate of 10 mV s⁻¹. (e) Sn mass-normalized CO₂RR activity. (f) Stability test of Cu₉₇Sn₃ at -0.75 V vs RHE.

Comment-4. Fig S9. I think capacitance measurements are not particularly accurate to assess the electroactive surface active area. I guess you assumed that the respective flat polycrystalline surfaces, i.e., with roughness factor 1, had similar capacitance constant value, and this is not necessarily true. But capacitance measurements is one of the most widely employed methodologies in literature to get an idea about the order of magnitude of the ECSA. I suggest the authors to add just a few references in which other researchers have used capacitance measurements for ECSA estimation in bi or multimetallic nanoparticles. In addition, I also suggest the authors to include the whole blank CVs in the S.I (i.e. between HER onset and onset of oxide formation) of Cu-Sn nanoparticles and Cu(100). Then, they should indicate which range of potential from the blank CV was selected to perform capacitance measurements. Sometimes, capacitance measurements could be conditioned by the potential limits.

Response: According to the reviewer's suggestion, we have added two references using capacitance measurements for ECSA evaluation of metallic nanoparticles (see Ref. 39,40).

Besides, the CVs of Cu₁₀₀, Cu₉₇Sn₃, and Cu₇₀Sn₃₀ have been added to indicate the range of capacitance measurements. Overall, the potential range is between the reduction and oxidation of Sn²⁺ and Sn (around -0.5 V vs SCE). The multi-scan CV curves (Figure S12d-f) show the typical near-square shape, indicative of negligible influence from oxidation/reduction reactions.

Supplementary Figure 12 | Electrochemically active surface area (ECSA) estimated from the double-layer capacitance. (a-c) CV curves for the identification of the capacitance measurements. (d-f) CV curves at different scan rates from 50 to 300 mV s⁻¹ for Cu₁₀₀, Cu₉₇Sn₃ and Cu₇₀Sn₃₀, respectively. (g) Double-layer capacitance of different electrodes. The fluctuation in CV curves can be attributed to the CO₂ gas bubbling during the tests and the Autolab setup is very sensitive to the current change. Note that the fitting error in (d) is very small, which suggests reliable double-layer capacitance calculation.

- [39] Li, C. W., Ciston, J. & Kanan, M. W. Electroreduction of carbon monoxide to liquid fuel on oxide-derived nanocrystalline copper. *Nature* **508**, 504-507 (2014).
- [40] Lai, Q., Yang, N. & Yuan, G. Highly efficient In-Sn alloy catalysts for electrochemical reduction of CO₂ to formate. *Electrochem. Commun.* **83**, 24-27 (2017).

Comment-5. Was the IR compensation the same in all the employed nanoparticulated catalyst surfaces?
Response: Yes, all LSV and potentiostatic data were corrected with an *iR* compensation of 80%, as indicated in the Electrochemical measurements section.

Kind regards,
 Chuan Zhao

REVIEWER COMMENTS

Reviewer #1 (Remarks to the Author):

The authors have provided more experimental data in the supporting document and rebuttal letter. In the main text, the STEM/EDX mapping (Fig. 1b) was not clear enough to show 'the Sn segregation' (I mentioned in the previous report). To justify this point, the authors have provided new HAADF-STEM and STEM/EDX mapping for the Cu₉₇Sn₃ nanoparticle. The HAADF-STEM image of the Cu₉₇Sn₃ nanoparticle in new Figure S1 is very similar to that of Cu₇₀Sn₃₀ in Fig. 1f. The shell thickness of Cu₉₇Sn₃ (new Fig.S1) is nearly the same as that of Cu₇₀Sn₃₀ (Fig. 1f). Furthermore, the authors mentioned in the rebuttal letter that surface content of Sn can reach ~8%. Sn has ten isotopes, seven of which have an abundance of more than 5%. These isotopes should be resolved clearly in the mass spectrum. However, the mass spectrum in new Figure S3 show very low Sn signals with incorrect ratios of the isotope abundance. Therefore, I am still not convinced that these peaks can be indexed as Sn.

Reviewer #2 (Remarks to the Author):

The authors have adequately addressed all my questions and remarks and I recommend the publication of this manuscript.

Reviewer #3 (Remarks to the Author):

I think the authors have properly addressed my comments. They have added more detailed discussion or description in the new manuscript, in relation to some important experimental points. Thereby I would recommend the paper for publication.

Response Letter

Nature Communications (NCOMMS-20-39314B)

Isolated copper-tin atomic interfaces tuning electrocatalytic CO₂ conversion

Wenhao Ren,^{1,†} Xin Tan,^{2,†} Jiangtao Qu,^{3,4} Sesi Li,⁵ Jiantao Li,⁶ Xin Liu,⁵ Simon P. Ringer,³ Julie M. Cairney,^{3,4} Kaixue Wang,⁵ Sean C. Smith² and Chuan Zhao^{1,*}

Response to Reviewer #1:

General Comment: The authors have provided more experimental data in the supporting document and rebuttal letter. In the main text, the STEM/EDX mapping (Fig. 1b) was not clear enough to show ‘the Sn segregation’ (I mentioned in the previous report). To justify this point, the authors have provided new HAADF-STEM and STEM/EDX mapping for the Cu₉₇Sn₃ nanoparticle. The HAADF-STEM image of the Cu₉₇Sn₃ nanoparticle in new Figure S1 is very similar to that of Cu₇₀Sn₃₀ in Fig. 1f. The shell thickness of Cu₉₇Sn₃ (new Fig.S1) is nearly the same as that of Cu₇₀Sn₃₀ (Fig. 1f). Furthermore, the authors mentioned in the rebuttal letter that surface content of Sn can reach ~8%. Sn has ten isotopes, seven of which have an abundance of more than 5%. These isotopes should be resolved clearly in the mass spectrum. However, the mass spectrum in new Figure S3 show very low Sn signals with incorrect ratios of the isotope abundance. Therefore, I am still not convinced that these peaks can be indexed as Sn.

Response: We very much appreciate Reviewer #1 for the time and insightful comments on our work. First, a set of new HRTEM and HAADF-STEM images have been obtained and added to the revised manuscript to give more structural information of Cu₉₇Sn₃ and Cu₇₀Sn₃₀ (see New Figure S1). It shows that there is a relatively thick amorphous SnO_x shell (2.0-3.8 nm) out of Cu nanoparticles on Cu₇₀Sn₃₀, while no obvious SnO_x shell is observed on Cu₉₇Sn₃. In addition to the TEM images that only studies very limited portions of the material, the XRD, UV-vis, XPS and XAS analysis give more statistical information on the overall structure, all of which suggest that the surface of Cu nanoparticles are fully covered by Sn in Cu₇₀Sn₃₀ but only partially covered in Cu₉₇Sn₃ (Figure S5,6). Besides, the overlapped EDX mapping of Cu₉₇Sn₃ and Cu₇₀Sn₃₀ derived from Fig. 1b,c are shown in Figure S2, where the surface segregation of Sn on both samples can be observed. The corresponding mass spectrum (Figure S2c,d) of Cu₉₇Sn₃ and Cu₇₀Sn₃₀ also exhibit similar peak ratios of Sn with the strongest signal at 3.44 keV (L_α of Sn), verifying that the presence of Sn in Cu₉₇Sn₃ rather than impurities or noise.

Second, the new mass to charge spectrum of APT with rebin-treatment to enhance the signal to background ratio is shown in Figure S4e,f.[1] It clearly shows that the mass spectrum of Sn matches very well to the corresponding natural abundance of main isotopes (blue lines). We believe this is a strong support for APT analysis, for which we thank the Reviewer for this insightful comment. Besides, according to the ICP result, the content of Sn is 2.7 at% in Cu₉₇Sn₃ (Table S1). Given the surface distribution of Sn atoms, it is reasonable to conclude that the surface content of Sn can reach up to ~8% from the APT analysis (Figure 2e).

To sum up, we have presented 4 experimental evidence together with theoretical calculations to complementary support the single-atom surface alloy structure. From experiment points of view: *i*) the atomic dispersion of Sn in $\text{Cu}_{97}\text{Sn}_3$ can be identified from the distributed bright spots in the HAADF-STEM images (Figure 1e and Figure S3); *ii*) the surface segregation of Sn on the Cu nanoparticle can be observed in the EDS mapping (Figure 1b,c and Figure S2a,b); *iii*) APT analyses reveal that the atomic isolation of Sn atoms on the surface of Cu nanoparticles from a three-dimensional perspective (Figure 2 and Supplementary Movie 01); *iv*) the Sn signals obtained in both EDS mapping (Figure S2c,d) and APT (Figure S4e,f) can be identified as Sn, rather than impurities or noise. From theoretical perspectives: *i*) the higher standard electrode potential of Cu^{2+} vs Sn^{2+} ($\text{Sn}^{2+} + 2e^- \rightleftharpoons \text{Sn}$, $E^0 = -0.13$ V vs. SHE; $\text{Cu}^{2+} + 2e^- \rightleftharpoons \text{Cu}$, $E^0 = 0.34$ V) results in a sequential reduction process, where the reduction of Cu^{2+} takes place first, followed by the deposition of Sn on Cu host surface. *ii*) DFT calculations suggest that Sn atoms tend to stay on the surface of Cu nanoparticles from an energetics perspective (Figure S18a). Based on the above analyses, we conclude that the $\text{Cu}_{97}\text{Sn}_3$ is a single-atom surface alloy, where the isolated Sn atoms are mainly anchored on the Cu host surface.

[1] Larson, D. J., Prosa, T., Ulfig, R. M., Geiser, B. P. & Kelly, T. F. Local electrode atom probe tomography. *New York, US: Springer Science* **2** (2013).

New Supplementary Figure 1 | TEM characterizations of $\text{Cu}_{97}\text{Sn}_3$ and $\text{Cu}_{70}\text{Sn}_{30}$. (a-d) The HRTEM and HAADF-STEM images of $\text{Cu}_{97}\text{Sn}_3$ with the spacing analysis of Cu lattice. (e-h) The

HRTEM images of $\text{Cu}_{70}\text{Sn}_{30}$ with the spacing analysis of Cu lattice.

New Supplementary Figure 2 | EDS analysis of $\text{Cu}_{97}\text{Sn}_3$ and $\text{Cu}_{70}\text{Sn}_{30}$. (a,b) The overlapped EDX mapping of $\text{Cu}_{97}\text{Sn}_3$ and $\text{Cu}_{70}\text{Sn}_{30}$ based on Fig. 1b,c, respectively. Cu (green) and Sn (red). (c,d) The corresponding mass spectrum analysis.

New Supplementary Figure 4 | Atom probe tomography analysis of $\text{Cu}_{97}\text{Sn}_3$ specimen. (a) Voltage curves. 3D tomography at the end of Stage 1 (b), Stage 2 (c), and Stage 3 (d). (e) The full-range mass to charge spectrum with rebin-treatment to enhance the signal to background ratio. (f) The detailed Sn isotopes analysis. Inset is the natural abundance of main isotopes of tin and the ratio is demonstrated as the blue lines, which is in good agreement with the experimental results.

Response to Reviewer #2:

General Comment: The authors have adequately addressed all my questions and remarks and I recommend the publication of this manuscript.

Response: We highly appreciate reviewer #2 for time and suggestions on our work.

Response to Reviewer #3:

General Comment: I think the authors have properly addressed my comments. They have added more detailed discussion or description in the new manuscript, in relation to some important experimental points. Thereby I would recommend the paper for publication.

Response: We very much appreciate reviewer #3 for the time and efforts on our work.

Kind regards,
Chuan Zhao

REVIEWERS' COMMENTS

Reviewer #1 (Remarks to the Author):

The authors have provided new results with proper analysis. I think they have addressed my concern. Therefore, I recommend the paper for publication.

Response Letter

Nature Communications (NCOMMS-20-39314B)

Isolated copper-tin atomic interfaces tuning electrocatalytic CO₂ conversion

Wenhao Ren,^{1,†} Xin Tan,^{2,†} Jiangtao Qu,^{3,4} Sesi Li,⁵ Jiantao Li,⁶ Xin Liu,⁵ Simon P. Ringer,³ Julie M. Cairney,^{3,4} Kaixue Wang,⁵ Sean C. Smith² and Chuan Zhao^{1,*}

Response to Reviewer #1:

The authors have provided new results with proper analysis. I think they have addressed my concern. Therefore, I recommend the paper for publication.

Response: We very much appreciate Reviewer #1 for the time and insightful comments on our work.

Kind regards,
Chuan Zhao